# Dynamic Distillation Network for Cross-Domain Few-Shot Recognition with Unlabeled Data

**Ashraful Islam**
Rensselaer Polytechnic Institute
islama6@rpi.edu

**Chun-Fu Chen**
MIT-IBM Watson AI Lab
chenrich@us.ibm.com

**Rameswar Panda**
MIT-IBM Watson AI Lab
rpanda@ibm.com

**Leonid Karlinsky**
IBM Research
leonidka@il.ibm.com

**Rogerio Feris**
IBM Research
rsferis@us.ibm.com

**Richard J. Radke**
Rensselaer Polytechnic Institute
rjradke@ecse.rpi.edu

## Abstract

Most existing works in few-shot learning rely on meta-learning the network on a large base dataset which is typically from the same domain as the target dataset. We tackle the problem of cross-domain few-shot learning where there is a large shift between the base and target domain. The problem of cross-domain few-shot recognition with unlabeled target data is largely unaddressed in the literature. STARTUP was the first method that tackles this problem using self-training. However, it uses a fixed teacher pretrained on a labeled base dataset to create soft labels for the unlabeled target samples. As the base dataset and unlabeled dataset are from different domains, projecting the target images in the class-domain of the base dataset with a fixed pretrained model might be sub-optimal. We propose a simple dynamic distillation-based approach to facilitate unlabeled images from the novel/base dataset. We impose consistency regularization by calculating predictions from the weakly-augmented versions of the unlabeled images from a teacher network and matching it with the strongly augmented versions of the same images from a student network. The parameters of the teacher network are updated as exponential moving average of the parameters of the student network. We show that the proposed network learns representation that can be easily adapted to the target domain even though it has not been trained with target-specific classes during the pretraining phase. Our model outperforms the current state-of-the art method by 4.4% for 1-shot and 3.6% for 5-shot classification in the BSCD-FSL benchmark, and also shows competitive performance on traditional in-domain few-shot learning task. Our code is available at: https://git.io/Jilgs.

## 1 Introduction

The tremendous success of deep learning in visual recognition tasks is, to a great extent, attributed to the availability of large scale labeled datasets. While humans can recognize an object by looking only at a few examples, modern deep neural networks require hundreds or thousands of images for each category to achieve human-level visual recognition capability. This has led to the research on few-shot learning which aims at learning from a much smaller dataset. In a typical few-shot learning setting, there are two stages: meta-training and meta-testing. In the meta-training stage, a base dataset with labeled images is provided to train the model. In the meta-testing stage, the learned model is quickly adapted to a set of novel classes with only a few examples per class (the support set) and evaluated on a set of test images from the same novel classes (the query set). The base classes and novel classes are typically disjoint, but the images are obtained from the same domain. However, in many real world settings, training the model on a base dataset from the same domain as the target dataset is difficult and infeasible. Guo et al. [7] proposed a cross-domain few-shot benchmark, BSCD-FSL, which contains datasets from extremely different domains. In this benchmark, the meta-training is

35th Conference on Neural Information Processing Systems (NeurIPS 2021).

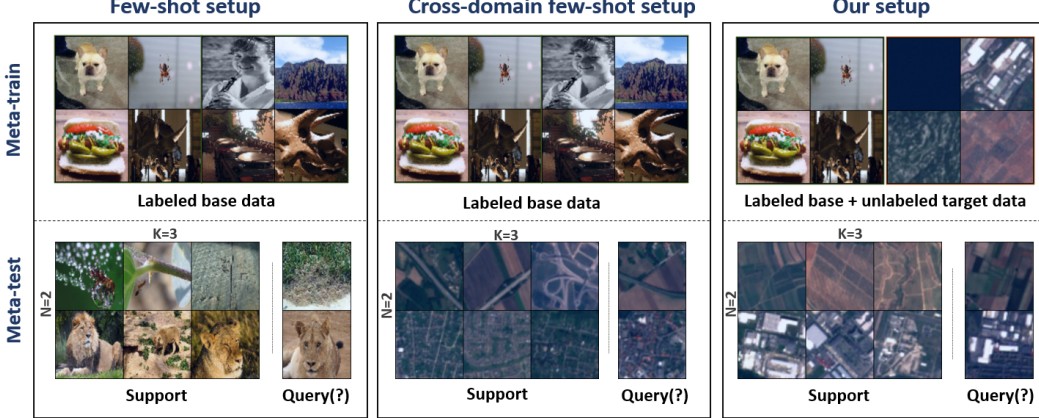

Figure 1: **Problem setup**. (Left) In typical few-shot learning task, a model is trained on a base dataset first during meta-training stage. In meta-testing stage, a few examples from novel classes, referred to as support set, are provided, and the network predicts the categories of different samples from the same classes as support set. The base dataset and the target dataset generally come from the same domain with disjoint categories. (Middle) In cross-domain few-shot learning, there is a domain gap between the base dataset and the target dataset. For example, in the figure, the base dataset contains natural images from miniImageNet [29], and the target dataset consists of satellite images from EuroSAT dataset [8]. (Right) Our setting is similar to cross-domain few-shot learning setup. However, additional unlabeled images are also available during meta-training stage. Although the unlabeled dataset comes from the same domain as the target dataset, it does not contain any images either from the support set or query set.

done on a labeled source dataset, and the few-shot evaluation is performed on a target dataset which is from different domain than the source dataset. The benchmark shows that traditional pretraining and finetuning outperforms more complicated meta-learning based few-shot learning methods by a significant margin.

In the real-world scenarios, the target domain should have many unlabeled images, and it might be beneficial to use the unlabeled data to learn more target domain specific representations. We hypothesize that using both labeled base data and unlabeled target data during training provides a common embedding for both base and target domain. Then the natural question could be - why not use the unlabeled target data only, it might provide more target-specific representation. One issue with this approach is that self-supervised learning generally requires a large amount of unlabeled data to work, and, as pointed out by Phoo and Hariharan [18], plain self-supervised learning struggles to outperform the naive transfer learning baseline in few-shot learning setup. Secondly, it has been shown that combining supervised and unsupervised learning during training provides more transferable representation [9]. We argue that similar conclusion holds for cross-domain few-shot learning, i.e., combining supervised and unsupervised loss provides better representation for the downstream task. Figure 1 illustrates our experimental setup in contrast to traditional few-shot learning or cross-domain few-shot learning setup. We show that labeled images from the base dataset are still important to learn generic image features, and images from the target domain, even if unlabeled, can help developing more target domain specific representations.

Figure 2 illustrates our approach. Our goal is to train a feature extractor which will be used to evaluate few-shot learning performance on the target dataset. We propose a dynamic distillation-based approach to this end. The student network consists of an encoder $f_s$ and classifier $g_s$, and the teacher network shares similar architecture as the student network (denoted as $f_t$ and $g_t$). The classifier $g_s$ is a linear layer that predicts the class-logits of the samples from the base dataset. We calculate a supervised cross-entropy loss between the student's predictions and ground-truth labels on the base dataset. For the unlabeled target data, we compute the teacher's prediction for a weakly-augmented version of an image and the student's prediction for a strongly augmented version of the same image, and optimize a distillation loss to match the predictions. We also apply sharpening in the teacher prediction to encourage low-entropy prediction from the student. Both the supervised loss and distillation loss are used to learn the student's weights. The teacher network is updated as a moving average of the student network. During few-shot evaluation, we only use the student

encoder $f_s$ as a feature extractor, learn a classifier head on the labeled support images consisting of few examples per category, and calculate the class predictions of the query images.

Our main contributions are:

- We propose a simple method for few-shot learning across extreme domain difference.
- We use dynamic distillation based approach that uses both labeled source data and unlabeled target data to learn a better representation for the few-shot evaluation on the target domain.
- Our method significantly outperforms the current state of the art in the BSCD-FSL benchmark with unlabeled images by **4.4**% for 1-shot and **3.6**% for 5-shot classification in terms of average top-1 accuracy. It even shows superior performance for in-domain few-shot classification on miniImageNet and tieredImageNet datasets.

## 2   Related Work

**Few-shot classification**    Few-shot learning methods can be divided into three broad categories - generative [36], metric-base [21, 28, 23] and adaptation-based [10, 14]. Early work on few-shot learning was based on meta-learning [29, 23, 10, 13]. Matching Networks [29] uses cosine similarities on feature vectors produced by independently learned feature extractors, while Relation Networks [23] learn its own similarity metric. MAML [10] learns good initialization parameters that can be quickly adapted to a new task. Prototypical Networks [21] learn a feature extractor that is used to calculate distances between features of test images and the mean features of support images. MetaOptNet [14] uses a discriminatively trained linear predictor to learn representations for few-shot learning.

**Self-training**    Self-training trains a student model that mimicks the predictions of a teacher model. Self-training can improves ImageNet classification [33]. It is also a dominant approach in semi-supervised learning, where the teacher network is used to create pseudo [35, 34] or soft labels [33] for a huge set of unlabeled images, and the student network is trained to mimic the teacher.

**Semi-supervised Learning**    Our method is inspired from recent developments in semi-supervised learning. Both [16] and [27] uses supervised cross-entropy loss with unsupervised regularization loss. Pseudo-labeling based approaches first train a model on a labeled dataset, use the trained model to create pseudo-labels of the unlabeled samples, and retrain the model with both labeled and pseudo-labeled samples [12, 1]. FixMatch [22] proposes a simplified model which simultaneously optimizes cross-entropy loss on the labeled samples and generates pseudo labels using the model's prediction on weakly-augmented unlabeled images. If the pseudo labels are confident enough, the model is trained to predict the pseudo labels with a strongly augmented version of the same images. We adopt a similar approach by imposing consistency regularization. However, while FixMatch is a semi-supervised technique where the unlabeled data is assumed to be from the same domain, our approach is applicable to the cross-domain few-shot learning problem. We also calculates the prediction from a mean teacher network instead of using the same network as FixMatch.

**Cross-domain few-shot learning**    Guo et al. [7] proposed a cross-domain few-shot learning benchmark, and noted that existing state-of-the-art approaches fail to achieve good accuracy on this benchmark. One potential solution could be to use an unlabeled dataset from the target to learn representations that are adaptable to a completely different domain. Many approaches also explored few-shot learning with unlabeled data [11, 15, 19]; however, most of these works still assume a smaller gap between the base and target domains. Our method shares some similarity with the recently developed STARTUP [18] method for cross-domain few-shot learning. STARTUP also uses unlabeled data for learning a better representation. However, STARTUP uses a fixed pretrained model to produce pseudo labels for the unlabeled samples, and then train the network with the labeled base dataset and pseudo-labeled target dataset. Additionally, STARTUP also incorporates a self-supervised contrastive loss on the unlabeled images to improve accuracy, where our method does not require additional contrastive loss. Actually, we argue that our distillation loss works like a self-supervised non-contrastive loss, similar to BYOL [6], for which we might not need to add any extra self-supervised loss. We propose a dynamic distillation approach, where the parameters of the teacher network are updated during training. We obtain the prediction for the weakly-augmented version of an unlabeled image from the teacher network, and optimizes the model such that the

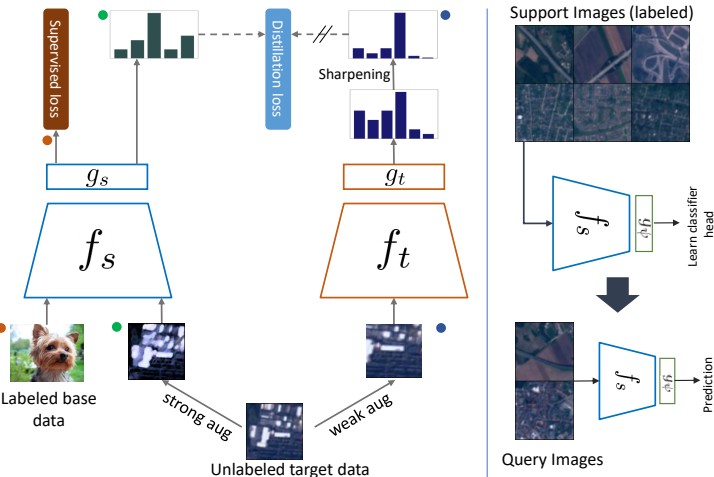

Figure 2: **Diagram of our approach**. Given labeled base data and unlabeled target data, our goal is to train a feature extractor which will be used to evaluate few-shot learning performance on the target dataset. The student network consists of an encoder $f_s$ and classifier $g_s$, and the teacher network share similar architecture as the student network. We use the labeled base dataset to optimize the supervised cross-entropy loss. For a target image, we compute the teacher's prediction for a weakly-augmented and student's prediction for a strongly augmented version of the image, and optimize the distillation loss to match the predictions. We also apply sharpening in the teacher prediction to encourage low-entropy prediction from the student. Both the supervised loss and distillation loss are used to learn student's weights. The teacher network is updated as a moving average of the student network. During few-shot evaluation, we simply learn a new classifier header on the few-shot support images, and evaluate on the query images.

prediction of the strongly augmented version of the same image obtained from the student network matches that of teacher network. Note that FixMatch [22] also uses similar consistency regularization loss for semi-supervised learning. To our knowledge, we are the first to use consistency regularization and dynamic distillation for cross-domain few-shot learning.

## 3 Methodology

### 3.1 Preliminary

**Few-shot Learning Formulation** A few-shot learning task consists of a support set $S$, which containing $K$ data points from $N$ classes for $N$-way $K$-shot task, and a query $Q = \{x_i\}_{i=1}^m$ consisting of data points only from the $N$ classes of the support set. The goal is to classify the query points with the help of the labeled support set. In the typical few-shot learning setting, (1) an embedding is learned from the base/source dataset $\mathcal{D}_S$, (2) a linear classifier is learned on top of the fixed embedding on the support set, and (3) the classifications of the query data points are determined.

**Cross-domain Few-shot Learning** The difference between the typical few-shot learning setup and cross-domain few-shot learning is that the base/source dataset is drawn from a very different domain than the target domain. Additionally, in our setting, we are provided unlabeled data points $\mathcal{D}_U = \{x_i\}_{i=1}^{N_U}$ from the target domain. The unlabeled dataset contains more classes than the support set. Given the base dataset $\mathcal{D}_S$, and an unlabeled set $\mathcal{D}_U$, we need to learn an embedding that can extract a representation that can be used for few-shot learning evaluation in the target-domain.

### 3.2 Proposed Method

**Encoder** We facilitate knowledge distillation to train our base encoder on both source datatset and target dataset. Denote the embedding network as $f_s$ that embeds an input image $x$ to a d-dimensional vector $f_s(x)$. We add a classifier header $g_s$ on top of $f_s$, which predicts $n_c$ logits from the embeddings, where $n_c$ is the total number of classes in the base dataset. Since the labels of the data points of the

base dataset are provided, we calculate the supervised cross-entropy loss:

$$l_{\text{CE}}(y, p) = H(y, p) \tag{1}$$

where $p = \texttt{Softmax}(g_s(f_s(x)))$, and $H(a, b) = -a \log b$.

**Dynamic distillation**    Additionally, we also have a teacher encoder $f_t$ and teacher classifier $g_t$. The task of the teacher network is to produce pseudo labels for the unlabeled images. Given an image $x_i$ from the unlabeled set $\mathcal{D}_U$, we compute the model's prediction $p_i^w$ from a weakly-augmented version (denoted as $x_i^w$) and $p_i^s$ from a strongly-augmented version (denoted as $x_i^s$) of the unlabeled image. The prediction of the weakly-augmented version is produced from the teacher network, which serves as a soft-target for the strongly-augmented images. We use the student network to get the prediction $p_i^s$ for the strongly-augmented images. Specifically,

$$p_i^s = \texttt{Softmax}(g_s(f_s(x_i^s))); \quad p_i^w = \texttt{Softmax}(g_t(f_t(x_i^w))/\tau) \tag{2}$$

where $\tau$ is a sharpening parameter. Note that we do not let gradient pass through the teacher network. We calculate the distillation loss by the cross-entropy function

$$l_U(p_i^w, p_i^s) = H(p_i^w, p_i^s) \tag{3}$$

Eq. 3 works like a consistency regularizer so that the network predicts similar scores for different augmented versions of the image. We can also consider Eq. 3 as self-supervised loss, similar to BYOL [6] or DINO [2]. However, one major difference is that - in BYOL or DINO the projection head is a random linear layer, where, we are using the supervised classification head as the projection head.

The total loss function is:

$$\mathcal{L} = \frac{1}{N_S} \sum_{(x_i, y_i) \in \mathcal{D}_S} l_{\text{CE}}(y, p) + \lambda \frac{1}{N_U} \sum_{x_i \in \mathcal{D}_U} l_U(p_i^w, p_i^s) \tag{4}$$

where $\lambda$ is a hyper-parameter. The loss function is used to update the parameters of the student network. For the teacher network, we use mean teacher approach [24], i.e., we update the teacher weights $\theta_t$ from the student weights $\theta_s$ by: $\theta_t = m\theta_t + (1 - m)\theta_s$, where $m$ is the momentum parameter. Note that when $m = 1$, we are essentially using fixed teacher, and when $m = 0$, the teacher and student share the same model. When $m > 0$, the teacher network is a moving average of the student network denoting the distillation process as dynamic. Please refer to the Appendix for PyTorch-like pseudo-code of our method.

## 4    Experiments

### 4.1    Experimental Setup

**Dataset**    We follow the evaluation protocol of the BSCD-FSL benchmark [7], which contains novel data from CropDisease [17], EuroSAT [8], ISIC [4], and ChestX [32]. The base dataset is miniImageNet [29], which contains 100 classes from ImageNet dataset [5] where each class has 600 images. The novel datasets are chosen based on increasing dissimilarity from the miniImageNet dataset. More details about the datasets are provided in the Appendix. Following [18], we randomly sample 20% of the data from each novel dataset to construct the unlabeled set $\mathcal{D}_U$, and the remaining images are used for evaluation, where we perform 5-way 1-shot and 5-way 5-shot classification. For evaluation metric, we report top-1 accuracy and 95% confidence interval over 600 runs. We also report evaluation results on the larger tieredImageNet dataset [19].

**Implementation details**    We use ResNet-10 as the backbone network [7, 18]. Our pretraining has two steps. In the first step, we train our network only on the miniImageNet dataset for 200 epochs. We use SGD with momentum 0.9, weight decay 1e-4, learning rate 0.01, batch size 32, and the cosine learning rate scheduler. In the next step, we use the miniImageNet-pretrained network, and use both the base dataset and the unlabeled dataset to optimize the loss function in Eq. 4 for 60 epochs. During training, we increase the weight of distillation loss $\lambda$ from 0 to 1 until 40 epochs

Table 1: **5-way 1-shot and 5-shot scores on the BSCD-FSL benchmark datasets**. The mean and 95% confidence interval of 600 runs are reported. The * indicates that the numbers are reported from [7] where no unlabeled data is used. The † are the numbers reported from [18], which uses 20% of the original set as the unlabeled dataset. We also use similar number of unlabeled images as [18]; however, the splits might be different for random sampling.

| Model | 1-shot | | | | 5-shot | | | |
|---|---|---|---|---|---|---|---|---|
| | EuroSAT | CropDisease | ISIC | ChestX | EuroSAT | CropDisease | ISIC | ChestX |
| MAML* | - | - | - | - | 71.70±.72 | 78.05±.70 | 40.13±.58 | 23.48±.48 |
| ProtoNet* | - | - | - | - | 73.29±.71 | 79.72±.79 | 39.57±.57 | 24.05±1.01 |
| MetaOpt* | - | - | - | - | 64.44±.73 | 68.41±.73 | 36.28±.50 | 22.53±.91 |
| STARTUP† | 63.88±.84 | 75.93±.80 | 32.66±.60 | 23.09±.43 | 82.29±.60 | 93.02±.45 | 47.22±.61 | 26.94±.45 |
| ProtoNet | 55.32±.88 | 52.94±.81 | 29.58±.57 | 21.32±.37 | 76.92±.67 | 81.84±.68 | 42.49±.58 | 24.72±.43 |
| MatchingNet | 54.88±.90 | 46.86±.88 | 27.37±.51 | 20.65±.29 | 68.00±.68 | 63.94±.84 | 33.96±.54 | 22.62±.36 |
| Transfer | 58.14±.83 | 68.78±.84 | 32.12±.59 | 22.60±.39 | 80.09±.61 | 89.79±.52 | 43.88±.57 | 26.51±.43 |
| SimCLR(Base) | 58.28±.90 | 61.58±.88 | 32.43±.56 | 22.37±.42 | 80.83±.64 | 83.44±.61 | 44.04±.55 | 26.63±.46 |
| SimCLR | 62.63±.87 | 69.22±.93 | 31.45±.59 | 23.59±.44 | 82.76±.59 | 89.31±.53 | 42.18±.54 | **29.56±.49** |
| STARTUP | 64.32±.88 | 74.45±.86 | 31.73±.57 | 22.27±.41 | 83.58±.60 | 92.41±.47 | 45.73±.62 | 26.21±.46 |
| Transfer+SimCLR | 63.91±.83 | 70.35±.85 | 31.67±.55 | **23.72±.44** | 85.78±.51 | 91.10±.49 | 45.97±.54 | 29.45±.10 |
| Ours | **73.14±.84** | **82.14±.78** | **34.66±.58** | 23.38±.43 | **89.07±.47** | **95.54±.38** | **49.36±.59** | 28.31±.46 |

using cosine scheduling. The sharpening temperature and teacher momentum parameter are set to 0.1 and 0.99 respectively. For the base images and weakly-augmented unlabeled images, we use the random-resize-crop, horizontal flip and normalization augmentations. For strong augmentation, we additionally use the color jitter, Gaussian blur, and random gray scale transformations. The other hyperparameters are kept the same. Refer to the supplementary for more details about hyper-parameter selection. For few-shot evaluation, we learn a logistic regression classifier on the support set, and evaluate on the query set.

## 4.2 Main Results

**Comparison to State-of-the-arts**   Table 1 shows the performance comparison of our approach with other methods on the BSCD-FSL benchmark. All models are trained on the miniImageNet dataset. "Transfer" denotes the baseline trained by cross-entropy loss on the base dataset. "SimCLR" is trained only on the unlabeled images. As noted by [9], self-supervised contrastive learning learns better transferable representation for a different downstream domain. Hence, we also create a "SimCLR(Base)" baseline that trains the encoder by optimizing self-supervised contrastive loss on the base dataset. "Transfer+SimCLR" refers to the model which is trained with supervised cross-entropy loss from the base dataset and self-supervised contrastive loss from the unlabeled target dataset, which has been reported to show superior transferability across domain [9]. STARTUP [18] is trained with three losses: cross-entropy loss on the base dataset, KL-divergence loss on the unlabeled dataset similar, and self-supervised contrastive loss on unlabeled images. STARTUP, Transfer+SimCLR and our approach use both the base dataset and additional unlabeled dataset during the representation learning phase.

Our method outperforms all meta-learning-based approaches by a significant margin at all settings. Moreover, compared to Transfer, we achieve more than ∼**5.5%** improvement for 5-shot classification on average. The performance improvement on 1-shot is more significant; we achieve **7.9%** improvement on average.

We outperform STARTUP by **5.5%** on EuroSAT, **3.1%** on CropDisease, **2.1%** on ChestX, and **3.6%** on ISIC for 5-way 5-shot classification. The performance improvement is also quite significant for 1-shot classification; specifically, our method achieves **8.8%** improvement on EuroSAT and **7.7%** improvement on CropDisease dataset over STARTUP. We only perform slightly worse on the ChestX dataset. For ChestX, it seems that pure unsupervised learning performs pretty well. Considering that our method does not use any self-supervised training or distillation, the performance improvement is impressive. Note that STARTUP uses a fixed teacher to extract pseudo-labels for the unlabeled images, whereas we extract pseudo labels from the weakly-augmented images from the same network that is being trained. In that sense, our model works like a dynamic teacher, where the pseudo labels get more refined as training progress. We hypothesize that the superior performance might be attributed to the dynamic approach of our model over STARTUP.

Table 2: **5-way 1-shot and 5-shot scores on the BSCD-FSL benchmark datasets when tieredImageNet is used as base dataset**. All models use ResNet-18 for backbone [25]. The mean and 95% confidence interval of 600 runs are reported. For EuroSAT and CropDisease dataset, our method achieves significant performance improvement over other models. The improvement for 1-shot learning is huge (over 7% for EuroSAT and 10% for CropDisease).

| | 1-shot | | | | 5-shot | | | |
| Model | EuroSAT | CropDisease | ISIC | ChestX | EuroSAT | CropDisease | ISIC | ChestX |
|---|---|---|---|---|---|---|---|---|
| Transfer | 58.07±.86 | 69.94±.87 | 29.76±.55 | 22.46±.41 | 81.34±.53 | 90.12±.49 | 41.27±.58 | 26.33±.45 |
| SimCLR(Base) | 62.14±.89 | 62.45±.90 | 31.03±.55 | 22.28±.40 | 81.85±.59 | 84.11±.60 | 42.91±.55 | 25.96±.44 |
| STARTUP | 64.32±.87 | 70.09±.86 | 29.73±.51 | 22.10±.40 | 85.19±.50 | 90.81±.49 | 43.55±.56 | 26.03±.44 |
| Transfer+SimCLR | 58.08±.83 | 71.25±.89 | 31.71±.55 | **23.81±.46** | 86.08±.47 | 91.31±.49 | 45.08±.56 | **30.26±.50** |
| Ours | **72.15±.75** | **84.41±.75** | **33.87±.56** | 22.70±.42 | **89.44±.42** | **95.90±.34** | **47.21±.56** | 27.67±.46 |

Table 3: **Few-shot evaluation on the same domain** in terms of 5-way 5-shot and 5-way 1-shot accuracy on miniImageNet and tieredImageNet datasets. We use ResNet-10 backbone for miniImageNet and ResNet-18 backbone for tieredImageNet.

| | miniImageNet | | tieredImageNet | |
| | 1-shot | 5-shot | 1-shot | 5-shot |
|---|---|---|---|---|
| ProtoNet | 51.06±.83 | 73.49±.63 | - | - |
| MatchingNet | 52.34±.81 | 67.28±.67 | - | - |
| Transfer | 53.40±.80 | 74.26±.64 | 58.61±.97 | 81.42±.65 |
| Transfer+SimCLR | 51.63±.82 | 74.65±.60 | 61.33±.96 | 82.89±.65 |
| STARTUP | 51.68±.84 | 74.05±.66 | 60.92±.96 | 82.11±.64 |
| Ours | **53.71±.83** | **76.02±.61** | **69.00±.96** | **85.93±.60** |

**Results with tieredImageNet base data**  tieredImageNet [19] is a subset of ImageNet dataset with 608 classes. The classes are grouped into 34 super-categories, from which 20 training categories (351 classes), 6 validation categories (97 classes), and 8 testing categories (160 classes) are selected. We use a larger backbone ResNet-18 for meta-training with tieredImageNet [25]. Table 2 shows the performance comparison with other models. We get similar conclusion as we get from Table 1. Particularly, we get 6.72% average improvement for 1-shot and 3.66% average improvement for 5-shot over STARTUP. However, we do not see much better accuracy than miniImageNet pretrained models even though we are using a much larger base dataset, which suggests that the *size of base dataset is not as important as other transfer learning task in cross domain few-shot learning*.

**Few-shot performance on similar domain**  It has been shown that self-training improves ImageNet classification [33]. Given the distillation approach of our model, one could expect that it might improve the few-shot accuracy even when the target data come from the same domain. Note that STARTUP does not improve the in-domain accuracy [18], even though it is also a self-training based model. We evaluate on miniImageNet and tieredImageNet dataset in terms of 5-way 1-shot and 5-way 5-shot performance. We use the official training split as base dataset, and the unlabeled target data are obtained from 20% of the novel (test) set and the rest are used for evaluation. For backbone, we use ResNet-10 for miniImageNet and ResNet-18 for tieredImageNet. Table 3 reports the results for in-domain few-shot performance. We see that our method achieves the best performance among the baselines. Particularly, for 1-shot learning in tieredImageNet, our model outperforms the best one by **7.7%**. We infer that our method can be safely applied to few-shot learning task when the domain gap between base and target dataset is small, which is in contrast with STARTUP that does not show improvement over "Transfer" for few-shot learning on similar domain.

## 4.3 Analysis

**Effect of dynamic distillation**  To understand how distillation helps to learn better representation, we use the pretrained models to extract features of the target dataset. Then we use KMeans algorithm to create clusters from the features. The number of clusters in the KMeans is set to be the number of classes of the target dataset. In Table

Table 4: **V-measure cluster score (%) [20] on the KMeans clustering of the extracted features with the ground-truth clustering**. The backbone is ResNet-10 pretrained on the miniImageNet dataset and/or the unlabeled target dataset.

| | EuroSAT | CropDisease | ISIC | ChestX |
|---|---|---|---|---|
| Transfer | 57.01 | 62.58 | **14.67** | 2.45 |
| SimCLR | 60.06 | 62.02 | 12.12 | **3.84** |
| STARTUP | 62.02 | 69.50 | 14.05 | 2.71 |
| Ours | **69.58** | **73.27** | 14.32 | 3.32 |

4, the V-measurement cluster scores [20] between the KMeans clusters and original ground-truth are shown. The V-score has 100% value when there is maximum agreement between ground-truth and predicted clusters, and 0% when there's no agreement. Table 4 shows that our method achieves higher v-scores for EuroSAT and CropDisease dataset, and the v-scores for ISIC and ChestX are also very competitive. It suggests that our model learns a good clustering of the target data even when we are not using any target labels. The clustering is much better when the domain gap is not extreme.

Fig. 3 shows t-SNE plots [26] from 10 representative classes from the CropDisease and EuroSAT datasets. We compare the embeddings extracted from "Transfer" and our approach. We see that our method creates better grouping on the embeddings of the target datasets, even though we do not use any labels for the target dataset during pretraining.

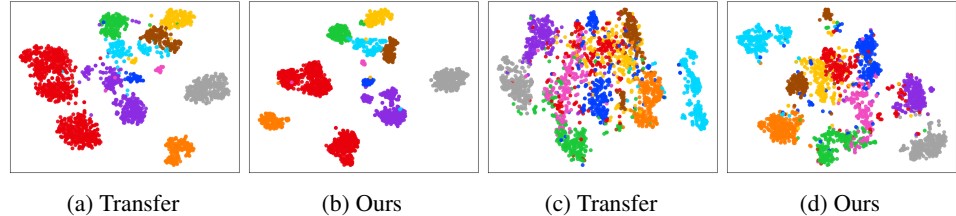

|  (a) Transfer | (b) Ours | (c) Transfer | (d) Ours |

Figure 3: t-SNE plot of 10 classes from CropDisease (a & b) and EuroSAT (c & d) test sets with features obtained from Transfer and our method.

**Comparison with self-supervised learning**   If we ignore the supervised loss, our model has similarity with self-supervised non-contrastive loss similar to BYOL or DINO. However, the projection head we are using to calculate the final predictions of the two different views of an unlabeled image is the same classification head that is used to predict the classification logits of the labeled base samples. In Table 5, "Ours (distillation head)" represents the model where we use separate projection head for the predictions of the unlabeled images. We see that separate projection head performs much worse. We found that the distillation loss is simply converging to a trivial solution in this case. To discourage trivial solution, we add recently developed tricks in self-supervised learning, namely, centering and strong augmentation - which turns the unlabeled branch similar to ResNet DINO [2]. "Ours (DINO head)" achieves better accuracy than "Ours (distillation head)", suggesting that it alleviates the issue of trivial solution. However, our original method still achieves a significant 3.07% more improvement. It is interesting to note that a separate projection head causes trivial solution for the unlabeled images, whereas our model does not converge to trivial solution with similar settings. We infer that *using a supervised classifier linear layer as the projection head can solve the issue of trivial solution for the self-supervised learning to some extent without requiring extra tricks like BatchNorm [6] or centering [2].* Additionally, it provides a better clustering of the unlabeled features, even if the unlabeled samples come from different domain than the labeled samples. Table 5 also shows results for "Ours + SimCLR", which simply adds a SimCLR loss for the unlabeled samples. It performs slightly better only in ChestX dataset. On average, the performance is similar to "Ours", which signifies that *there is no clear benefit using a self-supervised contrastive loss* to achieve better transferability for our method. Note that STARTUP comes to a different conclusion reporting that adding SimCLR loss consistently improves the performance.

Table 5: **Our method with self-supervised approaches**. The evaluation is performed on BSCD-FSL benchmark in terms of 5-way 5-shot accuracy (%). "Ours (distillation head)" refers to the model where we use a separate projection head for the distillation loss, which achieves much worse scores. In "Ours + DINO", we use a separate projection head with centering and strong augmentation as in DINO [2]. It achieves better performance than naive transfer, but still under-performs in comparison to our approach. "Ours+SimCLR" simply adds a self-supervised contrastive loss for the unlabeled samples. See Appendix for more details.

|                          | EuroSAT | CropDisease | ISIC  | ChestX |
|--------------------------|---------|-------------|-------|--------|
| Ours                     | 89.07   | 95.54       | 49.36 | 28.31  |
| Ours (distillation head) | 80.06   | 89.31       | 46.63 | 25.29  |
| Ours (DINO head)         | 85.74   | 90.55       | 46.24 | 25.42  |
| Ours + SimCLR            | 88.48   | 93.80       | 49.10 | 29.45  |

**Experiment of few-shot classification on fine-grained dataset**   CUB [30] contains 200 classes and 11,788 images of different bird species. 'miniImageNet->CUB' is an interesting experiment to show the transferability of different models to a fine-grained dataset. We report the results in Table 6 in terms of 5-way 5-shot scores. For CUB, we found that vanilla Transfer performs surprisingly well (also reported by [3]), and adding SimCLR with Transfer (Transfer+SimCLR) actually decreases the accuracy. Wallace and Hariharan [31] also experimented with different self-supervised methods on smaller domain and found that all of them under-perform for fine-grained task [2]. However, our method still performs the best, demonstrating the effectiveness of our approach in a fine-grained downstream task.

Table 6: **Experiment on mini-ImageNet -> CUB. All the scores are reproduced by us.**

| ProtoNet | Transfer | SimCLR | Transfer+SimCLR | STARTUP | Ours |
|----------|----------|--------|-----------------|---------|------|
| 63.19    | 68.72    | 62.84  | 67.82           | 66.10   | 69.50 |

**Experiment of distillation with unlabeled datasets from different domain**   Table 7 reports the few-shot accuracy when our model is trained on different unlabeled datasets. *The best accuracy is achieved when the unlabeled data and target data are from the same domain.* Even if the unlabeled data consists of images from multiple domains including the target domain (denoted as "Ours-all"), it still significantly under-performs the base model.

Table 7: **Effect of unlabeled datasets from a different domain than the target dataset** in terms of 5-way 5-shot accuracy (%). "Ours-X" denotes that we use base and "X" dataset during pretraining. "Ours-all" denotes that we use unlabeled images from all four target datasets.

|                  | EuroSAT | CropDisease | ChestX | ISIC  |
|------------------|---------|-------------|--------|-------|
| Ours-EuroSAT     | 89.07   | 90.43       | 26.02  | 46.82 |
| Ours-CropDisease | 81.86   | 95.54       | 26.17  | 45.12 |
| Ours-ISIC        | 81.94   | 89.69       | 26.70  | 49.36 |
| Ours-ChestX      | 81.87   | 90.38       | 28.31  | 45.20 |
| Ours-All         | 82.75   | 91.31       | 26.01  | 46.43 |

**Effect of data augmentation**   On the unlabeled images, we apply two types of augmentation: weak augmentation to extract pseudo labels and strong augmentation to impose consistency regularization. This setting is denoted as "weak-strong" (w-s), where 'weak' (w) augmentation is applied to the image that is fed into the teacher network and 'strong' (s) augmentation is applied to the image that is fed into the student network. We also show results with "weak-weak" (w-w), "strong-weak" (s-w) and "strong-strong" (s-s) augmentation settings in Figure 4a. Both "weak-weak" and "strong-strong" perform worse than the other augmentations. We also note that in self-supervised learning, generally strong augmentation is applied to all training images to get good performance. In our experiment, *we find that applying weak augmentation in one of the image pairs improves the performance*.

**More unlabeled data**   To measure the effect of amount of unlabeled dataset during pretraining, we divide the target dataset by 80% and 20% splits. The 20% split is used for evaluation. From the 80% split, we vary the amount of unlabeled data and then pretrain with source and the unlabeled set. Fig. 4b shows the average 5-shot accuracy for different amounts of the unlabeled dataset during the pretraining phase. As expected, *more information from the unlabeled dataset helps to learn better representations on the target domain*. However, the performance saturates later, and we get diminishing return for more unlabeled data. It also signifies that there are scopes to improve the performance by using more unlabeled data denoting potential future research direction.

### 4.4   Addition Ablation Studies

We perform several ablation studies of different components of our approach. All scores are reported for 5-way 5-shot evaluation. More ablations are provided in the Appendix.

**Longer training**   We use the base dataset pretrained network as initialization for our network, and then train on both base dataset and unlabeled dataset for 60 epochs. Figure 4c reports average 5-way 5-shot few-shot on the BSCD-FSL benchmark performance for our pretrained with longer training epoch. Training for more epochs can result in minor improvement (0.2%).

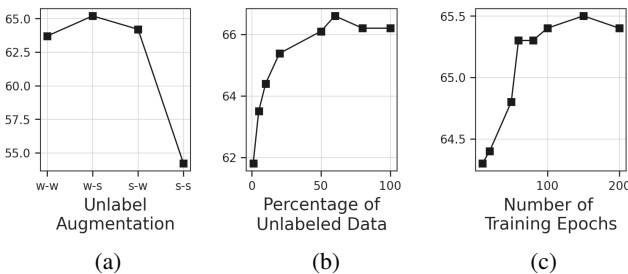

Figure 4: Ablation studies for (a) data augmentation on unlabeled images, (b) effect of longer training on base and unlabeled data, (c) amount of unlabeled data. The Y-axis represents average top-1 accuracy (%) on the four benchmark datasets for 5-shot classification for 600 episodes. We use miniImageNet for labeled source dataset and ResNet-10 as backbone.

Table 8: **Ablation studies on different settings**. Mean over 600 runs.

|  | EuroSAT | CropDisease | ISIC | ChestX |
|---|---|---|---|---|
| Ours | 89.07 | 95.54 | 49.36 | 28.31 |
| Ours(w/o base) | 82.11 | 90.52 | 39.76 | 26.83 |
| Ours(1-step) | 86.16 | 87.28 | 46.10 | 25.11 |

**Effect of base dataset**    Here, we perform experiments without the first term in Eq. 4, i.e., we train the network on the base dataset first and then re-train only on the unlabeled images (without joint training on the base dataset), denoted as "Ours (w/o base)". Table 8 shows that the performance of "Ours (w/o base)" is poor, suggesting that the *representation related to the labeled base dataset is still helpful to the target domain*.

**Training without pretrained model on base dataset**    We perform 2-step training during the representation learning phase - we first train the model on mini-IN only, and then jointly train on mini-IN and unlabeled images. In Table 8, "Ours(1-step)" denotes training on the unlabeled images and mini-IN from scratch for 300 epochs, which performs worse than 2-step training. Our assumption is that the *proposed model is also like self-training where a well-trained teacher is needed*.

## 5   Conclusion

We introduced a novel approach to utilize unlabeled data from the target domain for cross-domain few-shot learning. Experiments show that our method achieves state-of-the-art results in the BSCD-FSL benchmark for both 1-shot and 5-shot classification. Our model also outperforms other approaches in the same-domain few-shot learning. Future work can be focused on applying our approach in each episode during meta-testing so that the model can learn more category-specific representations.

## 6   Broader Impact

The approach tackles a practical problem of the existing few-shot learning setup that the base dataset and the novel samples generally come from different domain. Our work uses the unlabeled samples from the target dataset to learn more target specific representation. Like any other machine learning tool, the final impact depends on the intention of the people or institution applying it. However, it can be useful in drug discovery or medical image analysis where labeled dataset is limited.

## 7   Acknowledgments

This material is based upon work supported by the U.S. Department of Homeland Security, Science and Technology Directorate, Office of University Programs, under Grant Award 2013-ST-061-ED0001. The views and conclusions contained in this document are those of the authors and should not be interpreted as necessarily representing the official policies, either expressed or implied, of the U.S. Department of Homeland Security.

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
