# A  Appendix

## A.1  Description of evaluation datasets

For the evaluation dataset, we use four datasets from the BSCD-FSL benchmark [8]. **CropDisease** [19] contains natural images of diseased crop leaves categorized into 38 different classes. **EuroSAT** [9] is a satellite imagery dataset consisting of 27,000 labeled images with 10 different land use and land cover classes. **ChestX** [38] is comprised of X-Ray images, and **ISIC** [5] dataset contains dermoscopic images of skin lesions. Please refer to the BSCD-FSL paper [8] for more details about the dataset. Also note that we resized all images to 224x224 following [8].

We want to mention meta-dataset [31] which also aims at performing similar task as BSCD-FSL, however, meta-dataset is still mostly limited to natural images.

## A.2  Hyper-parameters

For training 'Transfer', we followed the protocol in [8]. STARTUP was trained using the parameters in the original paper [22]. For training the meta-learning based approach, we followed [3]. Note that we do not have proper validation set to tune hyperparameter on the target dataset, specially in 1-shot setting, therefore, we chose hyperparameters based on the performance on miniImageNet or tieredImageNet validation set. For the our method, we select learning rate 0.01 and batch size 32 as default pretraining parameters based on the accuracy on mini/tiered-ImageNet validation set.

For few-shot evaluation, we use logistic regression classifier on the extracted feature. It has been shown that simple logistic regression works best for few-shot learning instead of complicated meta-learning based strategy [30]. We adopt the same strategy for evaluating all the models.

## A.3  Pseudo-code

We show PyTorch-like algorithm of out approach in Algorithm 1.

---

**Algorithm 1** Pseudocode, PyTorch-like

---

```
# fs, ft: student and teacher backbone
# gs, gt: student and teacher classifier
# T: teacher temperature
# m: momentum rate to update teacher
ft.params = fs.params
gt.params = gs.params
for (xb, yb), xt in loader:
    # xb, yb: sample from base data
    # xt: sample from target data
    tb = gs(fs(xb)) # predicted logits
    loss_b = cross_entropy(tb, yb) # supervised loss

    xw, xs = weak_aug(xt), strong_aug(xt)
    tw, ts = gt(ft(xw)), gs(fs(xs))
    # sharpen + stop-grad
    tw = softmax(tw / T, dim=-1).detach()
    loss_t = cross_entropy(ts, tw) # distillation loss

    loss = loss_b + loss_t
    loss.backward() # back-propagate
    update(fs, gs) # update student

    # update teacher
    ft.params = m * ft.params + (1-m) * fs.params
    gt.params = m * gt.params + (1-m) * gs.params

def cross_entropy(t, y):
    # t: input logit
    # y: one-hot target
    t = softmax(t, dim=-1)
    return - (y * log(t)).sum(dim=-1).mean()
```

---

## A.4 Experiments

### A.4.1 Results with full-network finetuning

For the few-shot evaluation in the main paper, we use the backbone as fixed feature extractor following [22, 30]. In Table 9, we show results when we do full-network finetuning during few-shot evaluation, i.e., we finetune both the pretrained backbone and the classifier head for 5-way classification, on the BSCD-FSL datasets for miniImageNet pretrained model on ResNet-10 and tieredImageNet pretrained model on ResNet-18. We report both 1-shot and 5-shot top-1 accuracy. For hyperparameters, we followed similar settings as [8]. Our method performs perform the best in most cases. Note that for few-shot learning, just applying a logistic regression classifier on top of the fixed feature backbone performs better than full-network finetuning because of the size of the training dataset in the support set. This has also been pointed out by [30].

Table 9: **Results with full-network finetuning during few-shot evaluation.** 5-way 1-shot and 5-shot scores on the BSCD-FSL benchmark datasets on miniImageNet tieredImageNet dataset with full-network finetuning. The mean over 600 runs.

|  | EuroSAT | CropDisease | ISIC | ChestX |
|---|---|---|---|---|
| **5-way 1-shot** pretrained on **miniImageNet** | | | | |
| Transfer | 62.48 | 71.49 | 36.30 | 21.96 |
| SimCLR | 60.02 | 69.86 | 31.91 | 22.52 |
| STARTUP | 63.73 | 74.29 | 34.54 | 22.39 |
| Transfer+SimCLR | 66.78 | 74.01 | 36.39 | 24.79 |
| Ours | **68.94** | **80.42** | 36.19 | 23.60 |
| **5-way 5-shot** pretrained on **miniImageNet** | | | | |
| Transfer | 78.39 | 89.77 | 51.49 | 24.30 |
| SimCLR | 71.85 | 85.12 | 40.06 | 25.42 |
| STARTUP | 77.04 | 89.15 | 47.20 | 25.13 |
| Transfer+SimCLR | 82.24 | 89.50 | 48.27 | 28.63 |
| Ours | **84.31** | **93.71** | **49.17** | 27.71 |
| **5-way 1-shot** pretrained on **tieredImageNet** | | | | |
| Transfer | 60.38 | 67.56 | 35.70 | 22.96 |
| SimCLR | 57.54 | 62.50 | 32.73 | 23.17 |
| STARTUP | 65.33 | 71.54 | 33.31 | 23.23 |
| Transfer+SimCLR | 61.59 | 74.18 | 34.19 | 24.29 |
| Ours | **66.98** | **80.54** | **36.17** | **25.04** |
| **5-way 5-shot** pretrained on **tieredImageNet** | | | | |
| Transfer | 72.87 | 83.01 | 45.40 | 24.52 |
| SimCLR | 67.07 | 81.69 | 42.05 | 25.27 |
| STARTUP | 78.88 | 85.40 | 43.75 | 24.67 |
| Transfer+SimCLR | 81.44 | 89.04 | 45.59 | 28.31 |
| Ours | **83.41** | **93.35** | **49.31** | 27.06 |

## A.5 Comparison with additional self-supervised methods

We have performed additional 5-way 5-shot evaluation of BYOL, MoCo, Transfer+BYOL, and Transfer+MoCo, and report the results in the following table. BYOL and MoCo are trained on the unlabeled target images only, and Transfer+(BYOL/MoCo) is trained on both labeled base dataset (mini-ImageNet) and unlabeled target dataset. Similar to our comparison with SimCLR and Transfer+SimCLR in Table 10 in the main paper, our method outperforms all other models in all datasets except the ChestX dataset.

Table 10: **Comparison with additional self-supervised methods.** 5-way 1-shot and 5-shot scores on the BSCD-FSL benchmark datasets with models trained on miniImageNet.

|  | EuroSAT | CropDisease | ISIC | ChestX |
|---|---|---|---|---|
| 5-way **1-shot** pretrained on **miniImageNet** | | | | |
| BYOL | 82.95 | 91.52 | 41.22 | 26.44 |
| Transfer+BYOL | 85.59 | 89.83 | 45.57 | 29.10 |
| MoCo | 83.44 | 85.20 | 46.86 | 28.30 |
| Transfer+MoCo | 84.42 | 87.56 | 47.20 | 29.52 |

## A.6 Larger backbones

Table 11 reports few-shot results for miniImageNet-pretrained models using larger ResNet-18 backbone. Table 12 reports few-shot results for tieredImageNet-pretrained models using larger ResNet-34 backbone. We see that larger backbone does not necessarily perform better for CDFSL task, specially when we use smaller base dataset like miniImageNet. Similar results have also been observed by [22, 3].

Table 11: **Effect of larger backbone.** 5-way 1-shot and 5-shot scores on the BSCD-FSL benchmark datasets for **ResNet-18** backbone. The mean and 95% confidence interval of 600 runs are reported. **Larger backbone does not necessarily perform better for CDFSL task**.

|  | EuroSAT | CropDisease | ISIC | ChestX |
|---|---|---|---|---|
| 5-way **1-shot** | | | | |
| Transfer | 58.07±.86 | 69.94±.87 | 29.76±.55 | 22.46±.41 |
| STARTUP | 64.32±.87 | 70.09±.86 | 29.73±.51 | 22.10±.40 |
| Transfer+SimCLR | 58.08±.83 | 71.25±.89 | 31.71±.55 | 23.81±.46 |
| Ours | 72.15±.75 | 84.41±.75 | 33.87±.56 | 22.70±.42 |
| 5-way **5-shot** | | | | |
| Transfer | 79.66±.66 | 88.23±.55 | 45.37±.58 | 25.33±.44 |
| STARTUP | 84.88±.59 | 92.44±.47 | 46.58±.62 | 25.71±.44 |
| Transfer+SimCLR | 87.26±.47 | 91.07±.52 | 45.84±.54 | 29.89±.47 |
| Ours | 87.43±.52 | 92.23±.43 | 48.22±.59 | 26.62±.44 |

Table 12: **Effect of larger backbone.** 5-way 1-shot and 5-shot scores on the BSCD-FSL benchmark datasets for **ResNet-34** backbone pretrained on tieredImageNet dataset. The mean and 95% confidence interval of 600 runs are reported.

|  | EuroSAT | CropDisease | ISIC | ChestX |
|---|---|---|---|---|
| 5-way **1-shot** | | | | |
| Transfer | 57.83±.89 | 66.40±.89 | 28.66±.52 | 22.17±.41 |
| Ours | 72.14±.79 | 84.34±.74 | 33.99±.58 | 23.98±.44 |
| 5-way **5-shot** | | | | |
| Transfer | 81.44±.55 | 88.12±.53 | 40.07±.55 | 25.68±.43 |
| Ours | 90.16±.40 | 96.01±.33 | 47.50±.56 | 29.56±.49 |

## A.7 More ablations

Here, we perform more ablation analysis on different components of our approach. For the following experiments, we use ResNet-10 backbone, and miniImageNet base dataset. Evaluation is performed on the target dataset in terms of average 5-way 5-shot accuracy for 600 runs.

Table 13: **Effect of momentum parameter for teacher update.** 5-way 1-shot and 5-shot scores on the BSCD-FSL benchmark datasets for ResNet-10 backbone pretrained on miniImageNet dataset. The mean and 95% confidence interval of 600 runs are reported.

| | $m$ | EuroSAT | CropDisease | ISIC | ChestX |
|---|---|---|---|---|---|
| 5-way 1-shot | | | | | |
| Ours (fixed) | 0 | 69.99±.91 | 76.78±.81 | 35.99±.63 | 22.44±.43 |
| Ours (self) | 1 | 70.01±.87 | 82.27±.80 | 33.87±.59 | 22.98±.45 |
| Ours | 0.99 | **73.14±.84** | 82.14±.78 | **34.66±.58** | **23.38±.43** |
| 5-way 5-shot | | | | | |
| Ours (fixed) | 0 | 86.26±.53 | 93.24±.41 | 50.35±.60 | 26.56±.46 |
| Ours (self) | 1 | 88.17±.47 | 95.22±.37 | 48.45±.61 | 28.03±.47 |
| Ours | 0.99 | **89.07±.47** | **95.54±.38** | **49.36±.59** | **28.31±.46** |

**Effect of momentum parameter for teacher update**   Note that when teacher momentum parameter $m = 1$, we are essentially using fixed teacher, and when $m = 0$, the teacher and student share the same model. For our approach we use $m = 0.99$, however, we found that our method is not much sensitive to the value of $m$, specifically, it works pretty close for most values when $m > 0$. Table 13 shows the effect of the momentum parameter for updating the teacher network for fixed network($m = 0$), instance update($m = 1$), and momentum update($m = 0.99$).

**Evaluation from the teacher network**   In Table 14, "Ours (teacher)" denotes evaluation using the momentum teacher as the feature extractor. We see that momentum teacher does not perform well as a fixed feature extractor for CDFSL.

Table 14: **Ablation studies on different settings**. All the models are pretrained on miniImageNet dataset with unlabeled target data using ResNet-10 backbone. The evaluation is performed on the test dataset for 600 runs.

| | EuroSAT | CropDisease | ISIC | ChestX |
|---|---|---|---|---|
| Ours (teacher) | 81.67 | 90.32 | 45.83 | 26.84 |
| Ours (reset-head) | 88.37 | 95.32 | 48.82 | 28.11 |
| Ours (w/o distill schedule) | 88.56 | 95.72 | 48.00 | 28.25 |
| FixMatch | 86.15 | 94.12 | 26.62 | 48.51 |
| FixMatch (momentum) | 88.03 | 93.92 | 47.13 | 28.09 |

**Comparison with FixMatch**   Our method is inspired from FixMatch [26] which is a consistency based semi-supervised learning method. We also show performance of FixMatch-like model for CDFSL task. Note that FixMatch does not use momentum teacher and apply hard-thresholding for creating pseudo labels. Without the momentum teacher, the performance of "FixMatch" generally under-performs our method. If we use momentum teacher, denoted as "FixMatch (momentum)", the accuracy improves. It suggests that combining both momentum teacher and soft-pseudo-labelling is important for better performance.

**Is temperature sharpening necessary?**   We perform ablation on the sharpening temperature for the teacher network in Table 15. Note that $\tau = 1$ denotes no sharpening. The results suggest that lower sharpening temperature is better to learn good representation.

Table 15: **Ablation on sharpening temperature**.

| $\tau$ | EuroSAT | CropDisease | ISIC | ChestX | Mean |
|---|---|---|---|---|---|
| 0.02 | 88.44 | 95.25 | 49.28 | 28.25 | 65.30 |
| 0.06 | 88.56 | 95.46 | 48.23 | 28.17 | 65.11 |
| 0.2 | 88.40 | 95.18 | 47.99 | 28.17 | 64.94 |
| 0.5 | 88.97 | 95.09 | 48.12 | 28.35 | 65.13 |
| 0.8 | 88.08 | 94.70 | 49.11 | 27.87 | 64.94 |
| 1 | 88.06 | 94.83 | 49.83 | 28.08 | 65.20 |
| 2 | 86.30 | 90.49 | 47.06 | 26.92 | 62.69 |

## A.8 More dataset

We perform few-shot evaluation in 5 additional downstream datasets in Table 16. **DeepWeeds** [21] and **Flowers** [20] contain fine-grained natural images. **Resisc** [4] is a remote sensing image classification dataset. **Kaokore** [29] dataset contains 8848 face images from japanese illustration. **Omniglot** [13] contains 1623 different hand-writted characters from 50 different alphabets.

Table 16 shows similar conclusion that our approach is superior than other methods, particularly in 1-shot setting.

Table 16: **Few-shot evaluation on more downstream datasets**. Mean over 600 runs. miniImageNet base dataset.

|  | DeepWeeds | Kaokore | Flowers102 | Omniglot | Resisc45 | Mean |
|---|---|---|---|---|---|---|
| **5-way 1-shot** | | | | | | |
| ProtoNet | 33.62 | 27.66 | 54.47 | 69.62 | 44.97 | 46.07 |
| MatchingNet | 28.01 | 27.38 | 53.11 | 55.42 | 46.73 | 42.13 |
| Transfe | 38.88 | 31.65 | 65.51 | 82.25 | 55.43 | 54.74 |
| STARTUP | 37.93 | 31.71 | 64.94 | 86.97 | 54.03 | 55.12 |
| Transfer+SimCLR | 41.20 | 33.07 | 67.79 | 82.65 | 57.92 | 56.53 |
| Ours | 42.56 | 33.79 | 71.94 | 88.58 | 64.64 | 60.30 |
| **5-way 5-shot** | | | | | | |
| ProtoNet | 45.29 | 41.06 | 80.72 | 93.70 | 70.94 | 66.34 |
| MatchingNet | 36.65 | 37.38 | 70.62 | 64.43 | 64.61 | 54.74 |
| Transfe | 54.36 | 43.86 | 85.14 | 95.72 | 76.62 | 71.14 |
| STARTUP | 53.73 | 44.80 | 87.38 | 97.54 | 77.84 | 72.26 |
| Transfer+SimCLR | 59.81 | 48.10 | 89.32 | 97.13 | 81.50 | 75.17 |
| Ours | 61.44 | 48.45 | 90.16 | 97.83 | 84.15 | 76.41 |

### A.8.1 Additional Results

We show 5-way 20-shot and 50-shot performance on the miniImageNet pretrained models in Table 17.

Table 17: **5-way 20-shot and 50-shot scores on the BSCD-FSL benchmark datasets for ResNet-10 backbone**. The mean and 95% confidence interval of 600 runs are reported.

|  | EuroSAT | CropDisease | ISIC | ChestX |
|---|---|---|---|---|
| **5-way 20-shot** | | | | |
| ProtoNet | 82.53±.55 | 89.34±.48 | 50.79±.57 | 28.96±.43 |
| MatchingNet | 76.22±.57 | 76.66±.73 | 43.24±.53 | 26.17±.38 |
| Transfer | 88.70±.42 | 96.04±.26 | 56.96±.54 | 31.91±.47 |
| SimCLR | 89.62±.42 | 95.57±.30 | 50.96±.52 | 36.52±.51 |
| STARTUP | 90.34±.44 | 97.06±.24 | 56.99±.56 | 33.19±.46 |
| Transfer+SimCLR | 92.31±.33 | 96.70±.27 | 55.86±.52 | **37.51±.53** |
| Ours | **92.95±.33** | **98.07±.19** | **58.58±.57** | 35.89±.47 |
| **5-way 50-shot** | | | | |
| ProtoNet | 84.76±.51 | 91.35±.42 | 52.15±.53 | 31.34±.44 |
| MatchingNet | 43.37±.53 | 49.11±.66 | 28.61±.40 | 21.36±.29 |
| Transfer | 91.17±.36 | 97.59±.45 | 63.15±.39 | 35.35±.43 |
| SimCLR | 91.88±.33 | 97.28±.24 | 57.13±.42 | 40.26±.46 |
| STARTUP | 92.62±.47 | 98.33±.47 | 63.56±.42 | 35.67±.42 |
| Transfer+SimCLR | 93.92±.28 | 98.05±.21 | 61.40±.53 | **40.90±.41** |
| Ours | **94.38±.27** | **98.84±.15** | **63.82±.57** | 39.42±.40 |