# OpenReview forum: "Dynamic Distillation Network for Cross-Domain Few-Shot Recognition with Unlabeled Data"
_NeurIPS.cc/2021/Conference — NeurIPS 2021 Poster_

### Official Review · Reviewer_3Ucv · 2021-07-07

**Rating:** 7
**Confidence:** 4

**Summary:**

This paper tackles the cross-domain few-shot learning, by utilizing the unlabeled target data to encourage the feature extractor to learns the representations that is effective for target domain. They propose a dynamic distillation where the teacher is dynamically updated via exponential moving average during training. The dynamic distillation works similar to recent self-supervised learning method of BYOL, and successfully enhance the generalization capability of learned representation using unlabeled data. Dynamic distillation network achieves the state-of-the-art performance in BSCD-FSL benchmark.

**Limitations And Societal Impact:**

The authors addressed the limitations of their work.

**Main Review:**

[Strength]
1.	Dynamic distillation method utilizing the exponential moving average model as a teacher is technically sounds and interesting concept, considering the exponential moving average encoder or momentum encoder is widely used to generate the target feature in self-supervised learning, and successfully learns the more general representations.
2.	The authors provided various analysis and ablation experiment results, which is helpful to understand the proposed method. Analysis on the source of unlabeled data (Table 6), and effect of the data augmentation is interesting, not only from the point of view of cross-domain FSL but also from the point of view of self-supervised learning.
3.	The performance gain over prior work (STARTUP) is remarkable. Also, the authors provided various baselines related to transfer learning and self-supervised learning.


 [Question]

1.	In Tables 1 and 2, the performance of the proposed method is compared with the STARTUP and SimCLR-based baselines. However, BYOL is the most similar SSL method to the proposed method. How is the performance of the BYOL or Transfer + BYOL in cross-domain FSL?
2.	In ablation experiments, the authors presented the performance of Ours (w/o base), that pretrain the model on miniImageNet (base dataset) and then learn representation without labeled base dataset. What will happen if someone pretrains the model on base dataset, and then learn representation using the unlabeled images and images from base dataset (use the images only, and not use the label. Consider the images from base dataset as unlabeled data)? It can be an interesting experiment to figure out the effect of supervised loss in the proposed method.
3.	In line 212 to 214, the authors stated that the proposed method does not use any self-supervised training or distillation. However, the proposed method is dynamic distillation, and it utilizes the unlabeled samples like the self-supervised learning model, BYOL. Can you explain the exact meaning of the sentence in line 212 to 214?

* Typo: In line 136, "datatset" ->"dataset"

**Time Spent Reviewing:**

5

---

> ### Author Response · Authors · 2021-08-10
> **Response to Reviewer 3Ucv**
>
>
> We thank the reviewer for the positive and detailed review as well as the suggestions for improvement. Our response to the reviewer’s question is below:
>
> ***Q1: In Tables 1 and 2, the performance of the proposed method is compared with the STARTUP and SimCLR-based baselines. However, BYOL is the most similar SSL method to the proposed method. How is the performance of the **BYOL** or **Transfer + BYOL** in cross-domain FSL?***
>
> **A1**: We thank the reviewers for the suggestion. We performed additional experiments for *BYOL* and *Transfer+BYOL*, and report the 5-way 5-shot results below. Similar to our comparison with *SimCLR* and *Transfer+SimCLR* in Table 1 in the main paper, our method outperforms BYOL in all datasets except the ChestX dataset. We will include them in the camera-ready version.
>
>
> | Model           | EuroSAT | CropDisease | ISIC  | ChestX |
> | :-------------- | :------ | :---------- | :---- | :----- |
> | BYOL            | 82.95   | 91.52       | 41.22 | 26.44  |
> | Transfer + BYOL | 85.59   | 89.83       | 45.57 | 29.10  |
> | Ours            | 89.07   | 95.54       | 49.36 | 28.31  |
>
>
>
> ***Q2: In ablation experiments, the authors presented the performance of Ours (w/o base), that pretrain the model on miniImageNet (base dataset) and then learn representation without labeled base dataset. What will happen if someone pretrains the model on base dataset, and then learn representation using the unlabeled images and images from base dataset (use the images only, and not use the label. Consider the images from base dataset as unlabeled data)? It can be an interesting experiment to figure out the effect of supervised loss in the proposed method.***
>
> **A2**: Thank you for the suggestion, it is an interesting experiment that we didn't consider in the main paper. We show the 5-way 5-shot results of the suggested experiment (*Ours w/o base-labels*) below.
>
> | | EuroSAT | CropDisease | ISIC  | ChestX |
> | :------ | :------ | :---------- | :---- | :----- |
> | Ours w/o base-labels | 88.19   | 93.94       | 38.03 | 23.50  |
> | Ours            | 89.07   | 95.54       | 49.36 | 28.31  |
>
> Although the method works well for EuroSAT and CropDisease dataset, it performs poorly for ISIC and ChestX. It further clarifies the importance of the supervised loss in our method.
>
> ***Q3: In line 212 to 214, the authors stated that the proposed method does not use any self-supervised training or distillation. However, the proposed method is dynamic distillation, and it utilizes the unlabeled samples like the self-supervised learning model, BYOL. Can you explain the exact meaning of the sentence in line 212 to 214?***
>
> **A3**: Thanks for pointing this out. Line 212 is an unintentional typo from our side. We meant to say that "the proposed method does not use any supervision from the target domain during pretraining". We will change it in the final version.

---

> > ### Comment · Reviewer_3Ucv · 2021-08-23
> > **Post-rebuttal**
> >
> > I appreciate that the authors provided the additional experiments and clarification and improved my understanding on this paper.
> > I will maintain my score of 7.

---

> > > ### Author Response · Authors · 2021-08-30
> > > **Thank you**
> > >
> > > Thank you for your valuable comments and for appreciating our work.

---

### Official Review · Reviewer_pL1o · 2021-07-16

**Rating:** 4
**Confidence:** 4

**Summary:**

The author propose using teacher-student knowledge distillation for improving the performance of cross-domain few-shot learning.
Improved results on cross-domain few-shot learning benchmarks are observed and the performance on standard few-shot learning benchmarks are shown to be comparable to state-of-the-art.

**Main Review:**

The method introduced in this paper is well-presented.
The empirical results are somehow strong.

Concerns:
- My main concern to this paper is the motivation. Using student-teacher knowledge distillation is not a new idea in training deep neural networks. It remains unclear to me why such method is strong on cross-domain few-shot learning without seeing the experimental results. I expect the authors to provide more discussion on the advantages of the proposed methods, e.g., what specific natures of cross-domain few-shot learning is exploited? The current discussion shows only 'another way of using unlabeled data' in cross domain few-shot learning, yet this is something can be exploited in many tasks like standard few-shot learning, domain generalizations, etc..

- The paper is somehow poorly organized. It is surprising to see that the main discussion (methodology) is not even 1.5 page long with a Figure in it. I think this is also part of the reason that the motivation of this paper is not well presented.

- Some widely used cross-domain few-shot learning settings were set in [1], and are not included in the paper.

- The performance on standard few-shot classification datasets are actually not comparable to SOTA. E.g., according to the leaderboard [2], with the standard inductive setting, many methods can achieve over 54% with simple Conv-4 architecture on miniImageNet 5way 1shot. While in-domain few-shot classification is obviously less challenging, it is confusing to me why the proposed method performs poorly.

- Can the authors provide more discussion on how the proposed method is 'dynamic'?

[1] Chen, Wei-Yu, et al. "A closer look at few-shot classification." ICLR.
[2] https://few-shot.yyliu.net/miniimagenet.html

**Time Spent Reviewing:**

2

---

> ### Author Response · Authors · 2021-08-10
> **Response to Reviewer pL1o**
>
> We appreciate the review and thank the reviewer for the thoughtful feedback.
>
>
> ***Q1: My main concern to this paper is the motivation. Using student-teacher knowledge distillation is not a new idea in training deep neural networks. It remains unclear to me why such method is strong on cross-domain few-shot learning without seeing the experimental results. I expect the authors to provide more discussion on the advantages of the proposed methods, e.g., what specific natures of cross-domain few-shot learning is exploited? The current discussion shows only 'another way of using unlabeled data' in cross domain few-shot learning, yet this is something can be exploited in many tasks like standard few-shot learning, domain generalizations, etc..***
>
> **A1**: We thank the reviewer for the detailed comments. We provide several points below:
>
> 1. Although student-teacher knowledge distillation has been exploited in several computer vision problems, to the best of our knowledge, we are the first to apply this in the cross-domain few-shot learning problem.
> 2. We hypothesize that using both labeled base data and unlabeled target data during training provides a common embedding for both base and target domain. Then the natural question could be - why not use the unlabeled target data only, it might provide more target-specific representation. One issue with this approach is that self-supervised learning generally requires a large amount of unlabeled data to work. Secondly, it has been shown that combining supervised and unsupervised learning during training provides more transferable representation (*Islam et al., A Broad Study on the Transferability of Visual Representations with Contrastive Learning*). We argue that similar conclusion holds for cross-domain few-shot learning, i.e., combining supervised and unsupervised loss provides better representation for the downstream task.
> 3. An important aspect of our method is sharing the same head for supervised loss and distillation loss. Our distillation loss is similar to non-contrastive self-supervised loss like BYOL or DINO. However, it requires much more data and several training tricks to make the non-contrastive method like DINO to work. We argue that using the same distillation head for supervised loss resolves these issues. Please refer to line 262-283 in the main paper.
>
> 5. As for why our proposed method works on the cross-domain setting, we also refer to the "Effect of dynamic distillation" section of the main paper (line 242-261), where we show that our method creates better grouping on the embeddings of the target datasets even though we do not use any labels from the target dataset during pretraining.
> 6. We agree with the reviewer that this approach can be exploited in standard few-shot setting too, which we verified in "Few-shot performance on similar domain" section (line 228-240).
>
> ***Q2: The paper is somehow poorly organized. It is surprising to see that the main discussion (methodology) is not even 1.5 page long with a Figure in it. I think this is also part of the reason that the motivation of this paper is not well presented.***
>
> **A2**: Thanks for the suggestion. We will provide more details about the motivation of our proposed method in the final version.
>
> ***Q3. Some widely used cross-domain few-shot learning settings were set in [1], and are not included in the paper.***
>
> **A3**: For cross-domain few-shot evaluation, [1] uses only mini-ImageNet->CUB, i.e., training on mini-ImageNet dataset and evaluation on CUB dataset. We argue that this setting is rather limited as CUB contains only natural images like ImageNet. We adopt BSCD-FSL benchmark [6] which has a better distribution of downstream datasets from natural to medical images.
>
> [1] Chen, Wei-Yu, et al. "A closer look at few-shot classification." ICLR 2019.
>
> [6] Guo et al. A broader study of cross-domain few-shot learning. ECCV 2020.
>
>
> ***Q4: The performance on standard few-shot classification datasets are actually not comparable to SOTA. E.g., according to the [leaderboard](https://few-shot.yyliu.net/miniimagenet.html), with the standard inductive setting, many methods can achieve over 54% with simple Conv-4 architecture on miniImageNet 5way 1shot. While in-domain few-shot classification is obviously less challenging, it is confusing to me why the proposed method performs poorly.***
>
> **A4**: Thanks for the comment. In Table 3, we show the in-domain performance comparison with similar training and test set and similar evaluation protocol for the methods considered for cross-domain few-shot learning. First, we want to clarify that our goal is not meta-learning for in-domain few-shot evaluation. Our approach is about having a stronger pretraining if some unlabeled target-related data are available, which is not the evaluation protocol of the leaderboard. Moreover, as stated in lines 233-234, our method needs unlabeled data from novel classes, which results in the different test set for the evaluation than the leaderboard uses. Thus the results are not comparable.
>
>
> ***Q5: Can the authors provide more discussion on how the proposed method is 'dynamic'?***
>
> **A5**: The term 'dynamic' refers to the momentum teacher, as the parameters of the teacher network are updated during training from the parameters of the student network. We provided ablation on the importance of the momentum update in Table 11 in the supplementary material, which shows that we get around 1.47\% average improvement over fixed teacher for 5-way 5-shot evaluation.

---

> > ### Comment · Reviewer_pL1o · 2021-08-18
> > **Follow-up feedback**
> >
> > I sincerely appreciate the response provided by the authors. And I apologize for giving my further feedback late. As my opinion seems to disagree with other reviewers, I have spent additional hours on re-reading this paper and reconsider my comments.
> >
> > Here I have some further comments:
> >
> > **A1 Motivation**
> >
> > In my opinion, being the first to apply a technique to a new task cannot be considered as motivation or novelty, especially when the technique is a relatively established one and has been shown to be a general way of improving NN training.
> > So what I expect to see at this point is:
> > - Specific reasons that motivates you to apply this technique to this problem.
> > - Novel improvements you proposed that are tailored for this task based on your understandings and assumptions.
> >
> > Here are my comments to each point you provide:
> > 1. To me, applying knowledge distillation to a new task can hardly be considered as a contribution, as knowledge distillation has been proved as a strong method on improving representation learning. E.g., [1] has discussed the application of knowledge distillation + self-supervised learning on both standard and few-shot training.
> >
> > 2. What you provided can be part of the reason support the idea. However, I have following minor concerns:
> > - 'self-supervised learning generally requires a large amount of unlabeled data to work'. This argument needs further support. Please see [1,2,3] for a non-exhaustive list of applying self-supervised learning at few-shot scenarios.
> > - After reading the paper again, I have minor concern on the setting itself. As one of the main reasons we are studying cross-domain few-shot learning is that in practice, we have no idea where the model will be applied and how significant the domain discrepancy is. Assuming the availability of unlabeled target domain data can potentially restrict the applications to the known target domain only. In this case, a valid setting might be: training with domain A (labeled) and domain B (unlabled) and test on domain C (unavailable at training), which will have a more significant practical value. However, since this is an issue also shared by other work studying the same setting, failing to address this point will not effect the final score I recommend.
> >
> > 3. In paper, your discussion on this point is mainly based the results you got, but I'm asking motivation. If this improvement is really important, why it is not fully discussed in Section 1 and 3? And I'm having hard time understanding what the 'trivial solution' you were referring to in this discussion.
> >
> > 4. Again, what you provided is based on the results. And considering the general advantages of knowledge distillation on representation learning, I am not surprised by this result.
> >
> > 5. Please see my further comments on A4 below.
> >
> > **A2 Presentation**
> >
> > I'll reconsider the score I recommend if the authors can provide the plan on improving the presentation in more detail. Thanks.
> >
> > **A3 Other cross-domain few-shot setting**
> >
> > I respectfully disagree. The setting BSCD-FSL focusing on is more about sample-wise domain shift. While as ImageNet is a general classification dataset, and CUB is a fine-grain dataset, the source and target domains have dramatically different levels of inter-class discrepancy. miniImageNet -> CUB provides another perspective of cross-domain robustness measurement, therefore serves as a convincing benchmark and is not necessarily easier.
> >
> > Considering my late further comments, I will reconsider my score if the authors can provide additional results, but failing to provide them will not affect the score.
> >
> > **A4 Standard setting**
> >
> > I agree the settings are different. But since unlabeled data is introduced in training, will it be easier? To me, this setting is already closed to the 'transductive' setting recently attracts wide attention and achieves much higher performance comparing standard  'inductive' setting. Although not 100% the same since the unlabeled data is introduced in the testing stage in 'transductive' setting, but introducing more data in training stage should be able to ease the testing and make the numbers comparable, am I right?
> >
> > **A5 Dynamic**
> >
> > Thanks for your clarification. However, as also pointed by other reviewers, this 'momentum' idea has already been extensively studied in previous work in representation learning.
> >
> > ---
> >
> > I appreciate the hard work of authors. And please do not hesitate to correct me if I were wrong. However, given my further feedback, the score I recommend reminds at this point.
> >
> >
> > [1] Xu, Guodong, et al. "Knowledge distillation meets self-supervision." ECCV, 2020.
> >
> > [2] Chen, Da, et al. "Self-supervised learning for few-shot image classification." ICASSP, 2021.
> >
> > [3] Gidaris, Spyros, et al. "Boosting few-shot visual learning with self-supervision." ICCV, 2019.

---

> > > ### Author Response · Authors · 2021-08-30
> > > **Author Responses to Follow-up Feedback (Part 1)**
> > >
> > > Thanks for the feedback! We apologize for the delay. Please find below are our responses and let us know if you have any further questions/concerns.
> > >
> > > **A1 Motivation**
> > >
> > > **[Knowledge Distillation]** We agree with the reviewer that knowledge distillation has been proved to be a strong method for improving representation learning. However, most of the findings are applicable only for *in-domain data*. For example, [1] shows that KD enhances in-domain features for both linear transfer and few-shot learning, [4] uses KD for semi-supervised learning (where both labeled and unlabeled samples are from the same domain). Thus, it is still not clear whether the same conclusion holds for *cross-domain few-shot learning*. This mainly motivated us to exploit student-teacher knowledge distillation to tackle the problem of cross-domain few-shot learning where there is a large shift between the base and target domain. Moreover, our method is also different than [4] that uses separate layers for classifier head and self-supervised head, where we use the same classifier head as distillation head.  Without performing actual experiments and analysis, one could argue that reprojecting the features of target domain into a vastly different source domain (using the same head) might actually hurt the performance on the target domain. We believe our approach and analysis in the paper can be an important contribution to the existing literature and expand the applicability of knowledge distillation.
> > >
> > > [1] Xu, Guodong, et al. "Knowledge distillation meets self-supervision." ECCV, 2020.
> > >
> > > [4] Sohn, Kihyuk, et al. "Fixmatch: Simplifying semi-supervised learning with consistency and confidence."
> > >
> > >
> > > **[Self-Supervised Learning with Less Data]** We will make this point clear in the final version. Specifically, what we refer here is that self-supervised learning on the *unlabeled target data only* without using any source data is not optimal as self-supervised learning often requires a large amount of data to learn better features. We agree that self-supervised learning on limited data could help, but more data could help in learning more transferable representations.
> > >
> > >
> > > **[Problem Setup]** Thanks for the comment. We believe that the setup in our paper is practical. Imagine that a company collects a certain amount of unlabeled data and then can only afford to label few of them. In this case, the accessibility of target unlabeled data is available in the training time, and the few-labeled samples are used in the few-shot learning. More specifically, in our setup, we actually know where the model will be applied, but do not have many labeled samples from the target domain to train the network with supervised setting. The general paradigm in this setup is to train the network with self-supervised learning using the unlabeled target samples from a pretrained model. We propose a more effective method in this paper by using both the unlabeled target samples and labeled data from a different domain.
> > >
> > > The setup reviewer suggested (i.e., training with domain A (labeled) and domain B (unlabled) and test on domain C (unavailable at training)) is also interesting and we actually show few results on that in Table 6. Specifically, we used domains A and B to pretrain and then test on domain C. Nonetheless, the results suggest that directly using domain A and C leads the best results if the final evaluation is based on the domain C. In other words, the best accuracy is achieved when the unlabeled data and target data are from the same domain.
> > >
> > > **[Single Projection Head and Trivial Solution]** Sharing the same head for supervised and distillation loss leads to better performance because both source data and target data are projected to the same embedding space (single projection header), and the source data are actually used to regularize the embedding of the target data. Therefore, in order to discriminate among the source and target data in the same space including discrimination of each data point at the same time, the learned features must be more discrimiative. Then, it results in a much better feature representation. This is our main motivation why we would only use single header and it shows better results in comparison with separate projection headers. We will add this in the section 1 of the final version (see the revision plan described below).
> > >
> > >
> > > By trivial solution, we refer to the collapsed solution, i.e., outputting the same vector for all images, in non-contrastive self-supervised learning. Please check section 5.3 in [4], section 1 and 3 in [5], where the authors discuss why collapsed solution is a huge issue in non-contrastive self-supervised learning and how to avoid them with different tricks.  We hypothesize that using the same projection head for both supervised and self-supervised loss imposes a regularization effect which prevents collapse, without relying on the explicit (often complex) tricks mentioned in [4] and [5].
> > >
> > >
> > > [4] Caron, Mathilde, et al. "Emerging properties in self-supervised vision transformers."
> > >
> > > [5] Grill, Jean-Bastien, et al. "Bootstrap your own latent: A new approach to self-supervised learning."
> > >
> > >
> > > **A2 Presentation**
> > >
> > > Below we provide a detailed plan on how we will improve the presentation to make our motivation and setup clear in the final version.
> > >
> > > - In the second paragraph of 'Introduction', we will add an additional discussion (about 2-3 lines) on why our setup is practical and what's the difference between usual cross-domain few-shot and our setup (see our response above on Problem Setup).
> > > - We will discuss the motivation of our approach in the third paragraph of section 1 (before illustrating our approach). For motivation, we will first mention why using both labeled source data and unlabeled target data during pre-training can be helpful (see our initial response A1-2). Then we will mention how using the same projection head could lead to better performance (see our updated response above on Single Projection Head and Trivial Solution). We anticipate this should be around 5-6 lines.
> > > - We will move some parts of the 4th paragraph in the original paper (similarity and difference with recent methods) to section 2.
> > > - In section 4.3, we will add additional experiments for miniImage->CUB as a separate subsection to prove the effectiveness of our approach to fine-grained domain where self-supervised methods underperform (see response below).
> > >
> > > We hope the above plan for reorganization of the paper will clarify our motivation earlier in the Introduction section. Please let us know if you have any further suggestion on this. We would be more than happy to incorporate them in the final version.
> > >
> > >
> > > **A3 Other cross-domain few-shot setting**
> > >
> > > **[Experiment on miniImageNet -> CUB]** Thanks for suggesting this experiment. We agree with the reviewer that mini-ImageNet->CUB is an interesting experiment to show the transferability of different models to a fine-grained dataset. We perform additional experiments for mini-ImageNet->CUB, and report the results (5-way 5-shot) below:
> > >
> > >
> > > | Model | CUB |
> > > | -------- | -------- |
> > > | MatchingNet     | 58.23 |
> > > | ProtoNet     | 63.19  |
> > > | Transfer     | 68.72  |
> > > | SimCLR     | 62.84  |
> > > | Transfer+SimCLR     | 67.82  |
> > > | STARTUP     | 66.10  |
> > > | Ours     | **69.50**  |
> > >
> > > Note that we did not directly cite the number in [1] as the test setup is different, all the results are reproduced by us. For CUB, we found that vanilla `Transfer` performs surprisingly well (also reported by [1]), and adding SimCLR with Transfer (`Transfer+SimCLR`) actually decreases the accuracy. Wallace et al. also experimented with different self-supervised methods on smaller domain and found that all of them underperform for fine-grained task [2]. However, *our method still performs the best, demonstrating the effectiveness of our approach in a fine-grained downstream task*. We will add this finding as a separate subsection in Section 4.3 in the main paper.
> > >
> > >
> > > [1] Chen, Wei-Yu, et al. "A closer look at few-shot classification." ICLR 2019.
> > >
> > > [2] Wallace, Bram, and Bharath Hariharan. "Extending and analyzing self-supervised learning across domains."

---

> > > > ### Author Response · Authors · 2021-08-30
> > > > **Author Responses to Follow-up Feedback (Part 2)**
> > > >
> > > > **A4 Standard setting**
> > > >
> > > > **[Inductive vs Transductive Setup]** As rightly stated by the reviewer, our setting is not entirely same to the *transductive setting* as unlabeled data is not being used in testing stage of our framework. On the other hand, it is also slightly different from *inductive setting* as our method needs unlabeled data from novel classes, which results in a different test set for evaluation compared to the inductive methods in the leaderboard.
> > > >
> > > > Despite different settings, our results are very much comparable to the previous findings on inductive methods. For example with the same ResNet-10 backbone, the `Baseline` method in [1] achieves an accuracy of 74.69 and 52.37, while our`Transfer` baseline obtains 74.26 and 53.40 for 5-shot and 1-shot evauation on miniImageNet respectively. We compare the results of other methods with the `Transfer` baseline in Table 3 of the main paper, where we achieve the best performance for in-domain evaluation (`Ours`: 76.02 for 5-shot and 53.71 for 1-shot on miniImageNet). We also note that the improvement of in-domain evaluation for tieredImageNet dataset is pretty significant over the `Transfer` baseline, and even comparable to the [leaderboard (tieredImageNet)](https://few-shot.yyliu.net/tieredimagenet.html) methods, implying that our approach can achieve better performance for in-domain few-shot evaluation with more data. Overall, we infer that our approach provides a good backbone/feature-extractor even for in-domain few-shot evaluation.
> > > >
> > > > Moreover, most methods in the [leaderboard (miniImageNet)](https://few-shot.yyliu.net/miniimagenet.html) apply different post-processing networks/tricks with the backbone. For example, LFT [3] integrates additional transformation layers into the encoder backbone, and FEAT [2] uses additional layers and tricks including set-to-set function and Bidirectional LSTM/Transformer layer to achieve good results. We argue that our method is not directly comparable with these approaches, as we are simply using a pretrained backbone as a feature extractor for the few-shot evaluation without any post-processing layers or tricks. However, one can still use our trained backbone as initialization for other methods. To illustrate this, we perform an experiment where we use ResNet-10 backbone trained with `Transfer` baseline and ResNet-10 backbone trained with our method for two different initialization of FEAT. We train with the default hyperparameters in FEAT for 200 epochs and evaluate on a subset of the mini-ImageNet test images that were not used in the pretraining of our approach. We show the 5-way 1-shot results below along with the original accuracy of the models used for initialization:
> > > >
> > > > | Method  | Accuracy (1-shot 5-way) |
> > > > | -------- | -------- |
> > > > | Transfer |  53.40 |
> > > > | Ours     |  53.71 |
> > > > | FEAT (initialization with Transfer) |  54.66 |
> > > > | FEAT (initialization with Ours)     |  56.70 |
> > > >
> > > > Indeed, pretrained backbone from our method can be used as a better initializer for FEAT than that of transfer baseline, which signifies the efficacy of our approach over baseline models.
> > > >
> > > > [1] Chen, Wei-Yu, et al. "A closer look at few-shot classification." ICLR 2019.
> > > >
> > > > [2] Ye, Han-Jia, et al. "Few-shot learning via embedding adaptation with set-to-set functions."
> > > >
> > > > [3] Tseng, Hung-Yu, et al. "Cross-domain few-shot classification via learned feature-wise transformation."

---

> > > > > ### Author Response · Authors · 2021-09-02
> > > > > **Request for feedback**
> > > > >
> > > > > Dear Reviewer pL1o,
> > > > >
> > > > > Thank you for your constructive comments and suggestions. As the discussion phase is nearing its end, we wondered if you might still have any concerns that we could address. We believe our response addressed all your questions/concerns, and hope that the work's impact and results are better highlighted with our responses.
> > > > >
> > > > > We appreciate all your feedback and hope our response helps your final recommendation. Thank you!
> > > > >
> > > > > Best wishes,
> > > > >
> > > > > Authors

---

### Official Review · Reviewer_zcQX · 2021-07-16

**Rating:** 6
**Confidence:** 4

**Summary:**

The submission attempts to address the cross-domain few-shot learning problem by utilizing dynamic distillation from additional unlabeled target data. It simplifies the STARTUP framework by removing the contrastive loss and using a similar network for teacher and student. Moreover, it introduces different data augmentations to learn better representations. The teacher network is also trained dynamically to enhance the training of the student network.

**Limitations And Societal Impact:**

The proposed approach doesn't achieve the best performance on ChestX data.

**Main Review:**

Strength:
+ The submission is well-written and easy to follow.
+ Comprehensive experiments and promising results.
+ Detailed analysis and convincing ablation study.

Weakness:
- Limited novelty since the approach combines the mean teacher approach from [23] and self-supervised loss from [5] and [2], yet achieving good performance in the cross-domain few-shot learning context.
- A question about the distillation loss l_u. The cross-entropy function cannot guarantee that the prediction of the student is equal to the prediction of the teacher. Please clarify this.
- It would be better to explain why and how to choose the weak and strong data augmentations.

Minor suggestions:
- Please highlight the numbers in Tables 5 and 6.


**Time Spent Reviewing:**

5

---

> ### Author Response · Authors · 2021-08-10
> **Response to Reviewer zcQX**
>
> We thank the reviewer for the insightful comments. Below we address the concerns.
>
> ***Q1: Limited novelty since the approach combines the mean teacher approach from [23] and self-supervised loss from [5] and [2], yet achieving good performance in the cross-domain few-shot learning context.***
>
> **A1**: To clarify, there are crucial differences between our methods and other self-supervised methods like [2, 5]. The projection head we are using to calculate the final predictions of the two different views of an unlabeled image is the same classification head that is used to predict the classification logits of the labeled base samples. This is important as we show in Table 5 in the main paper that separate projection head performs much worse. We refer to the section "Comparison with self-supervised learning" (line 262-283) for detailed comparison with the other self-supervised learning methods. We also agree that our teacher-student approach is inspired from [23]. However, the other details are important (e.g., using the same classification head for distillation loss, augmentation, two-step training) to make it work on the cross-domain few-shot setting.
>
> [2] M. Caron, H. Touvron, I. Misra, H. Jégou, J. Mairal, P. Bojanowski, and A. Joulin. Emerging properties in self-supervised vision transformers.
>
> [5] J.-B. Grill, F. Strub, F. Altché, C. Tallec, P. H. Richemond, E. Buchatskaya, C. Doersch, B. A. Pires, Z. D. Guo, M. G. Azar, et al. Bootstrap your own latent: A new approach to self-supervised learning.
>
> [23] A. Tarvainen and H. Valpola. Mean teachers are better role models: Weight-averaged consistency targets improve semi-supervised deep learning results.
>
>
> ***Q2: The cross-entropy function in the distillation loss \\( l_u \\) cannot guarantee that the prediction of the student is equal to the prediction of the teacher. Please clarify this.***
>
> **A2**: To clarify, the cross entropy function \\( l_u(p^w, p^s) = H(p^w, p^s) \\) is similar to KL divergence \\( D_{KL}(p^w, p^s) = H(p^w, p^s) - H(p^w) \\) from the perspective of optimization for \\( p^s \\), as \\( p^w \\) is held constant with *stop-grad* operation. Hence, \\( l_u(p^w, p^s) \\) will be minimized when the distributions \\( p^w \\) and \\( p^s \\) are the same.
>
>
> ***Q3: Why and how to choose the weak and strong data augmentations?***
>
> **A3**: For weak augmentation, we use random-resize-crop, horizontal flip and normalization, and for strong augmentation we additionally use additionally use the color jitter, Gaussian blur, and random gray scale transformations (line 183-186 in the main paper).
>
> To answer why we chose weak and strong augmentation, we provided ablations in the main paper (line 289-297). We infer that *weakly-augmented images provide better pseudo-labels* from the teacher network such that the student network can be optimized to make the outputs from strongly-augmented images to be consistent with the outputs from the weakly-augmented images.

---

> > ### Comment · Reviewer_zcQX · 2021-08-25
> > **Thank you for the rebuttal.**
> >
> > I still have concerns here:
> >
> > Q3: Maybe I asked the question in a wrong way. How did the authors build up the sets of weak/strong augmentation? Have you ever tried different components for the two sets of data augmentation?

---

> > > ### Author Response · Authors · 2021-08-26
> > > **Response will be available soon**
> > >
> > > Dear Reviewer,
> > >
> > >
> > > Thanks for your feedback. We are still preparing the response, and it will be available by 8/29.
> > >
> > >
> > > Best Wishes,
> > >
> > > Authors

---

> > > > ### Author Response · Authors · 2021-08-30
> > > > **Clarification on Data Augmentation**
> > > >
> > > > We thank the reviewer for the feedback and apologize for giving our followup response late. Below we provide more details about the augmentations and will add a discussion on this in our final version.
> > > >
> > > > For weak augmentation, we adopt the standard augmentation used in ResNet training [2], i.e., random-resize-crop, and horizontal flip. On the other hand, for strong augmentation, we follow the augmentations used in [1, 5], i.e., additionally use the color jitter, Gaussian blur, and random gray scale transformations on top of random-resize-crop, and horizontal flip.
> > > >
> > > > We have also tried other augmentations, but did not find them perform well. Specifically, we tried 'multi-crop' [3] and 'rand-aug' [4], both of which can be considered stronger than the original one used as part of the *strong* augmentation. We report the results below which show that too strong augmentation does not help much for cross-domain transfer. It also depends on the target domain. For instance, 'multi-crop' works slightly better on CropDisease dataset, but performs poorly on ISIC. The default augmentations that we used in the main paper generally perform well across domains.
> > > >
> > > > | Model           | EuroSAT | CropDisease | ISIC  | ChestX |
> > > > | :-------------- | :------ | :---------- | :---- | :----- |
> > > > | Ours (multi-crop)   | 88.25 | **95.98** | 41.83 | 25.42 |
> > > > | Ours (rand-aug)      | 84.58 | 92.91 | 47.41 | 25.03 |
> > > > | Ours | **89.07**   | 95.54       | **49.36** | **28.31**  |
> > > >
> > > >
> > > > [1] Ting Chen et al., "Big Self-Supervised Models are Strong Semi-Supervised Learners".
> > > >
> > > > [2] Kaiming He et al., "Deep Residual Learning for Image Recognition".
> > > >
> > > > [3] Caron, Mathilde, et al. "Unsupervised learning of visual features by contrasting cluster assignments."
> > > >
> > > > [4] Cubuk, Ekin D., et al. "Randaugment: Practical automated data augmentation with a reduced search space."
> > > >
> > > > [5] Chen, Xinlei, et al. "Improved baselines with momentum contrastive learning.

---

> > > > > ### Comment · Reviewer_zcQX · 2021-08-30
> > > > > **Post-rebuttal**
> > > > >
> > > > > Thank you for the discussion. I will keep my score of 6.

---

> > > > > > ### Author Response · Authors · 2021-08-30
> > > > > > **Thank you**
> > > > > >
> > > > > > Thank you for your valuable comments and feedbacks.

---

### Official Review · Reviewer_eC3H · 2021-07-16

**Rating:** 6
**Confidence:** 4

**Summary:**

The paper proposes a method for training on “cross-domain few-shot recognition”. This is a task in which the goal is to transfer representations learned on a base, labeled dataset into a target few-shot dataset that is of a different domain. In particular, the paper focuses on the setting where unlabeled data is available for the target domain.

The method proposed comprises of training a “student model” on the base data in a supervised manner, while also utilizing a distillation loss on the target data, using a “teacher network”’s predictions as the objective. The teacher is set as an exponential moving average of the student’s parameters. After this initial training phase, the student gets a new randomly initialized head to learn on the few-shot labeled target data.


**Limitations And Societal Impact:**

Yes

**Main Review:**

The method proposed is very interesting and seems to yield good results on the BSCD-FSL benchmark. I found the insights in lines 42-44, 266-276 particularly insightful.

# Major Comments
It is very related to many similar techniques, such as MoCo, SimCLR, BYOL, etc. In particular, it seems like it differs from MoCo only by the addition of a supervised head and the downstream finetuning on few-shot datasets. I found it disappointing to not see it in the baselines compared. Other baselines (line 111) seem to have been left out because they “assume smaller gap [between domains]”, but this means that this claim isn’t shown empirically. Overall, this decreases my confidence in the high performance of the proposed method.

I also am not convinced by the conclusion of lines 226-227, as another possible explanation is that the target tasks are simply too easy for extra large data to show any benefit.

# Minor Comments
I found it surprising that training with Ours-All in Table 6 did not yield the best results in all target tasks. Why do the authors think that is?

Overall I found the paper very interesting and well explained, and would consider raising my score if my main concerns are addressed.


**Time Spent Reviewing:**

2

---

> ### Author Response · Authors · 2021-08-10
> **Response to Reviewer eC3H**
>
>
> Thank you for the insightful comments. Below we address the concerns.
>
> ***Q1: Comparison with other methods like BYOL, MoCo, etc.***
>
>
> **A1**: We have performed additional 5-way 5-shot evaluation of *BYOL*, *MoCo*, *Transfer+BYOL*, and *Transfer+MoCo*, and report the results in the following table. *BYOL* and *MoCo* are trained on the unlabeled target images only, and *Transfer+(BYOL/MoCo)* is trained on both labeled base dataset (mini-ImageNet) and unlabeled target dataset. Similar to our comparison with *SimCLR* and *Transfer+SimCLR* in Table 1 in the main paper, our method outperforms all other models in all datasets except the ChestX dataset. We will include them in the camera-ready version.
>
> | Model           | EuroSAT | CropDisease | ISIC  | ChestX |
> | :-------------- | :------ | :---------- | :---- | :----- |
> | BYOL            | 82.95   | 91.52       | 41.22 | 26.44  |
> | Transfer + BYOL | 85.59   | 89.83       | 45.57 | 29.10  |
> | MoCo            | 83.44   | 85.20       | 46.86 | 28.30  |
> | Transfer + MoCo | 84.42   | 87.56       | 47.20 | **29.52**  |
> | Ours           | **89.07**| **95.54**   | **49.36** | 28.31  |
>
>
> ***Q2:  Compariosn with other in-domain semi-supervised few-shot learning baselines.***
>
> **A2:** Thank you for pointing this out. We show additional results for 5-way 5-shot evaluation below for semi-supervised soft k-Means Prototypical Network from [18], which uses both labeled base dataset and unlabeled target dataset to create class prototypes. Apart from the CropdDisease dataset, the results are worse than simple ProtoNet. It suggests that soft k-means ProtoNet is not optimal for cross-domain few-shot setting where the unlabeled samples are obtained from a different domain than the base dataset.
>
> | Model                 | EuroSAT | CropDisease | ISIC  | ChestX |
> | :-------------------- | :------ | :---------- | :---- | :----- |
> | ProtoNet              | 76.92   | 81.84       | 42.49 | 24.72  |
> | Soft k-Means ProtoNet | 72.10   | 82.43       | 41.44 | 24.26  |
> | Ours                  | 89.07   | 95.54       | 49.36 | 28.31  |
>
> Note that, [14] uses transductive inference that classifies the entire test set at once, hence it's not directly applicable with our evaluation protocol.
>
>
> [14] Y. Liu, J. Lee, M. Park, S. Kim, E. Yang, S. J. Hwang, and Y. Yang. Learning to propagate labels: Transductive propagation network for few-shot learning.
>
> [18] M. Ren, E. Triantafillou, S. Ravi, J. Snell, K. Swersky, J. B. Tenenbaum, H. Larochelle, and R. S. Zemel. Meta-learning for semi-supervised few-shot classification
>
> ***Q3: I also am not convinced by the conclusion of lines 226-227, as another possible explanation is that the target tasks are simply too easy for extra large data to show any benefit."***?
>
> **A3:** Thanks for the excellent suggestion. We agree that this could be another explanation, and will update it in the camera ready version.
>
> ***Q4: I found it surprising that training with Ours-All in Table 6 did not yield the best results in all target tasks. Why do the authors think that is?***
>
> **A4:** Excellent question! It has been shown in the literature that not every unlabeled sample is equally helpful for the downstream task in self-supervised or semi-supervised learning [1]. Our experiment suggests that unlabeled samples from a completely different domain than the downstream task might actually hurt the performance.
>
> [1] Zhongzheng Ren, Raymond A. Yeh, Alexander G. Schwing. Not All Unlabeled Data are Equal: Learning to Weight Data in Semi-supervised Learning.  NeurIPS'20.

---

> > ### Comment · Reviewer_eC3H · 2021-08-20
> > **re: rebuttal**
> >
> > Post-rebuttal comments: I thank the authors for the additional experiments and the rebuttal. I maintain my score of 6

---

> > > ### Author Response · Authors · 2021-08-30
> > > **Thank you**
> > >
> > > Thank you for your thoughtful feedbacks and comments.

---

### Decision · Program_Chairs · 2021-09-27

**Decision:**

Accept (Poster)

**Comment:**

The paper received mixed ratings, with three reviewers recommending acceptance and one rejection. The reviewers' main concerns include the novelty of the method, its motivation, missing comparison to some baselines, and the clarity of some parts of the paper. The paper went through several rounds of questions/answers, and the authors feedback addressed the concerns of most reviewers. The most negative reviewer did not update their score, but we believe that the latest feedback from the authors, on which the reviewer did not comment, better addressed some of the concerns raised by the reviewer. As such, we believe that the paper can be accepted to NeurIPS but strongly recommend the authors to incorporate their feedback in the final version.